# Content and form of original research articles in general major medical journals

Nicole Heßler[1], Andreas Ziegler [2,3,4,5,6]*

1 Institut für Medizinische Biometrie und Statistik (IMBS), Universität zu Lübeck, Universitätsklinikum-Schleswig-Holstein, Campus Lübeck, Lübeck, Germany, 2 Cardio-CARE, Medizincampus Davos, Davos, Switzerland, 3 School of Mathematics, Statistics and Computer Science, University of KwaZulu Natal, Pietermaritzburg, South Africa, 4 Department of Cardiology, University Heart and Vascular Center Hamburg, University Medical Center Hamburg-Eppendorf, Hamburg, Germany, 5 Centre for Population Health Innovation (POINT), University Heart and Vascular Center Hamburg, University Medical Center Hamburg-Eppendorf, Hamburg, Germany, 6 Swiss Institute of Bioinformatics, Lausanne, Switzerland

* ziegler.lit@mailbox.org

**Data Availability Statement:** All relevant data are within the paper and its Supporting Information files.

**Funding:** The authors received no specific funding for this work.

## Abstract

The title of an article is the main entrance for reading the full article. The aim of our work therefore is to examine differences of title content and form between original research articles and its changes over time. Using PubMed we examined title properties of 500 randomly chosen original research articles published in the general major medical journals BMJ, JAMA, Lancet, NEJM and PLOS Medicine between 2011 and 2020. Articles were manually evaluated with two independent raters. To analyze differences between journals and changes over time, we performed random effect meta-analyses and logistic regression models. Mentioning of results, providing any quantitative or semi-quantitative information, using a declarative title, a dash or a question mark were rarely used in the title in all considered journals. The use of a subtitle, methods-related items, such as mentioning of methods, clinical context or treatment increased over time (all p < 0.05), while the use of phrasal tiles decreased over time (p = 0.044). Not a single NEJM title contained a study name, while the Lancet had the highest usage of it (45%). The use of study names increased over time (per year odds ratio: 1.13 (95% CI: [1.03–1.24]), p = 0.008). Investigating title content and form was time-consuming because some criteria could only be adequately evaluated by hand. Title content changed over time and differed substantially between the five major medical journals. Authors are advised to carefully study titles of journal articles in their target journal prior to manuscript submission.

## Introduction

Researchers have the duty to make the results of their research on human subjects publicly available according to the declaration of Helsinki [1], and many recommendations for the reporting of studies have been developed. An overview on these reporting guidelines is provided by the EQUATOR (Enhancing the QUAlity and Transparency Of health Research) network, which aims to tackle the problems of poor reporting [2]. One consequence of systematic reporting is that many scientific articles are organized in the same way [3, 4], and they

**Competing interests:** There are no patents, products in development or marketed products to declare. AZ is a licensed Tim Albert trainer. Tim Albert trainings deal with advising people how to write medical papers. AZ has held several courses in the past based on Albert's concept. This does not alter our adherence to PLOS ONE policies on sharing data and material.

generally follow the IMRAD structure, which stands for Introduction, Methods, Results, And Discussion. The IMRAD structure is also standard for the writing of abstracts. It is therefore of interest to researchers how they can individualize their reports to increase the citation counts, which is one important measure for career advancement [5].

Approximately 30 factors affecting citation frequency have already been identified [6–9]. While journal- and author related factors are generally not modifiable, some article-specific factors are subject to active modification by the authors. Especially the title has been proposed as a modifiable component of a research article [9–11]. Researchers should use titles that accurately reflect the content of their work and allow others easily to find and re-use their research [12]. Most research has focused on the form of article titles because these analyses could be performed automatically and are not very time-consuming [9, 13, 14].

While the article content has been studied well both in features, such as tense, voice and personal pronouns, and in the IMRAD sections between different research disciplines, title content has received less attention, and the main focus was title length [15, 16]. One reason could be the lack of automated internet searches until approximately 25 years ago. For example, PubMed was first released in 1996, Web of Science is online since 1997 and Google Scholar started not earlier than in 2004. With the advent of automated internet-based searches the importance of the title has changed, and it is now the "billboard" of a research article [17]. Another reason could be that these evaluations have to be made manually, and they are thus time-consuming [18]. An additional time-consuming factor could be that guidelines such as the Standards for Reporting of Diagnostic Accuracy (STARD) statement [19] strongly recommend that at least two observers should do an independent evaluation where applicable.

Most articles investigating the form of the title compared whether the title was a full sentence [20], descriptive, indicative, or a question [18, 21], or whether the title included non-alphanumeric characters, such as a colon or dash [22]. Very few publications looked at other title components of a research article. Specifically, Kerans, Marshall [23] compared the frequency of Methods mentioning or Results mentioning for the general major medical journals, specifically the New England Journal of Medicine (NEJM), the BMJ, the Journal of the American Medical Association (JAMA), and the Lancet by analyzing the first approximately 60 articles published either in 2015 or 2017 in each of the journals. Both articles investigated only a few months from a single publication year per journal. The development of title content over time was thus not considered.

The aim of our work therefore was to examine properties of title content for original research articles published in one of the five major clinical journals (BMJ, JAMA, Lancet, NEJM, and PLOS Medicine (PLOS)) over the 10-year period from 2011 until 2020. Specifically, we aimed at identifying differences between the five journals and changes over time regarding title content and title form. We also compared our findings to those of Kerans et al. [15, 23].

## Materials and methods

### Search in Medline and Web of Science

The search strategy has been described in detail elsewhere [9]. In brief, we first extracted all original research articles finally published between 2011 and 2020 in the five major clinical journals BMJ, JAMA, Lancet, NEJM, and PLOS. The restriction to the publication year 2011 allows for proper comparisons between journals because PLOS was reshaped in 2009 [24].

The variables PubMed identifier (PMID), journal name, article title, author names, publication year, citation, PubMed Central identifier (PMCID) and digital object identifier (DOI) were extracted from the Medline search. From the Web of Science, we reduced available

information to journal name, article title, PMID, abstract for the identification of original research articles, DOI and publication date. Both PMID and DOI were used to merge articles identified in Medline (n = 8396) and the Web of Science (n = 10267). Articles being listed with an abstract remained in the data set, while articles only listed in the Web of Science were excluded. Articles being only downloaded in the Medline files were checked whether they were indeed original research articles. If not, they were excluded as well. After data cleaning, a set of 8096 articles was available.

## Evaluation of title content and form

To investigate title content and form, we randomly selected 500 original articles from the years 2011 to 2020. The random selection was done with stratification by journal and year so that ten original articles per year (100 articles per journal) were randomly chosen. To avoid a priori information on the specific journal article, only the title and the PMID were presented in the database. In addition, the order of the 500 articles was randomized prior to evaluation. All article titles were evaluated manually by two raters/authors. Both raters performed a training and independently evaluated 25 randomly selected journal articles—five per journal—prior to the evaluation of the 500 articles. These training articles were excluded from the main evaluation. Conflicts in ratings were solved by agreement.

Items for title content and form are displayed in **Table 1** and were inspired by other works [15, 25, 26]. One reviewer asked for the discoverability in each of the title items, therefore, we provided two examples of article titles with the result of our evaluation in **Table 1**.

The first block of **Table 1** reports results on title content. Title content was divided into the topics Methods and Results. The former is concerned with the mentioning of methods in the title, such as the study design or a novel technique used in the paper [15]. Other elements from the methods concern the mentioning of a patient population, the geography, the clinical context, an intervention, and the use of study names in the title. The latter examines results mentioned in the manuscript. The first question was whether results were stated in the title at all. More detailed were the questions whether quantitative information or semiquantitative or ordinal information was provided [26]. It was also noted whether the title reported on a relation between two or more variables [26].

The second block of **Table 1** is related to the form of a title divided into the topics Methods, and Conclusion/Discussion. The use of abbreviations, dashes and subtitles was investigated for the Methods. The three single items for Conclusion/Discussion were whether the title was declarative, phrasal, or formulated as a question.

Recently, we performed an analysis after an automatic search for country and city mentioning in the title by the use of the R package `maps` [9], and we did not expect substantial differences to our hand search.

## Sample size considerations

The main aim of our work was to investigate trends over time by a regression model. In general, regression models have a sufficient sample for a single independent variable, such as time, if n ≈ 50 [27, 28]. Specifically, for a weak effect size of $R^2 = 0.14$ [29], the required sample size is 51. In case of a weak effect size of Cohen's f [29] with $f^2 = R^2 / (1 - R^2) = 0.14$, the required sample size is 403 to achieve a power of 80%. A sample size of 500 as used in our work yields a power of 87.75% at a significance level of 5%.

## Statistics

Descriptive statistics for the specified title properties, i.e., absolute and relative frequencies were reported for each journal over time, refraining of descriptive p-values for investigating

**Table 1. Items for title content and form.**

| Title | Topic | Variable | Item | Title example 1 | Title example 2 |
|---|---|---|---|---|---|
| | | | | **"Raltegravir-intensified initial antiretroviral therapy in advanced HIV disease in Africa: A randomised controlled trial"** | **"APP, PSEN1, and PSEN2 mutations in early-onset Alzheimer disease: A genetic screening study of familial and sporadic cases"** |
| Content | Methods | Methods mention | Methods are mentioned, such as the study design. This included mentioning of the study design or the type of analysis. | Yes | Yes |
| | | | | "randomised controlled trial" | "genetic screening study" |
| | | | E.g. *"randomized controlled trial"*, *"efficacy"*, *"trend"* or *"cost-effectiveness analysis"* | | |
| | | Patient population | A patient population is named. | Yes | Yes |
| | | | E.g. *"patients with acute traumatic brain injury* or *people with dementia"* | "advanced HIV disease in Africa" | "early-onset Alzheimer disease" |
| | | Geography | The geographic location is named. | Yes | No |
| | | | E.g. *"England"* or *"English"* | "Africa" | |
| | | Clinical context | The clinical context is indicated. | Yes | Yes |
| | | | E.g. *"acute traumatic brain injury* or *dementia"* | "advanced HIV disease" | "mutations in early-onset Alzheimer disease" |
| | | Intervention | An intervention is named. | Yes | No |
| | | | E.g. *"HPV vaccination* or *oral corticosteroids"* | "Raltegravir-intensified initial antiretroviral therapy" | |
| | | Study name | The title contains a study name for the work presented in the paper. | No | No |
| | | | E.g. *"VENUS randomized clinical trials"* or *"French GAZEL prospective cohorts"* | | |
| | Results | Results mention | Results are mentioned. | No | No |
| | | | E.g. *"reduced risk of Plasmodium vivax malaria"* | | |
| | | Quantitative information | The title contains results as quantitative information (specific value). | No | No |
| | | | E.g. *"doubled risk"* or *"HR of 1.42"* | | |
| | | Semiquantitative information | The title contains results as semiquantitative or ordinal information. | No | No |
| | | | E.g. *"increased"*, *"decreased"*, *"high"* or *"low"* | | |
| | | Relation | A relation between variables is mentioned. | No | No |
| | | | E.g. *"association"*, *"change"*, *"correlation"*, *"determinants"*, *"effect"*, *"evidence"*, *"heterogeneity"*, *"impact"*, *"influence"*, *"maximize"*, *"Mendelian randomization"*, *"outcomes"*, *"pattern"*, *"predictors"*, *"relation"*, *"remission"*, *"risk"*, *"trend"*, *"variability"* or *"variation"* | | |
| Form | Methods | Abbreviation | An abbreviation is used. | Yes | Yes |
| | | | E.g. *"COPD"*, *"HIV"*, *"MRI"*, *"PSA"* or *"U.S"* | "HIV" | "APP, PSEN1, PSEN2" |
| | | Dash | The title contains a dash. | No | No |
| | | | E.g. *"Consequences of undervaccination–measles outbreak."* | | |
| | | Subtitle | A subtitle is used. | Yes | Yes |
| | | | E.g. *"alcohol consumption and fecundability: prospective Danish cohort study"* | "Raltegravir-intensified initial antiretroviral therapy in advanced HIV disease in Africa: A randomised controlled trial" | "APP, PSEN1, and PSEN2 mutations in early-onset Alzheimer disease: A genetic screening study of familial and sporadic cases"" |
| | Discussion/ Conclusion | Declarative title | The title is declarative (= full-sentence structure). | No | No |
| | | | E.g. *"Doxycycline reduces scar thickness"* | | |
| | | Phrasal title | The title is phrasal (no full-sentence structure but containing any form of a verb except active verbs). | No | No |
| | | | E.g. *"prolonged survival"* or *"estimated mortality on HIV"* | | |
| | | Question | The title contains a question. | No | No |
| | | | E.g. *"Is food insecurity associated with HIV risk?"* | | |

journal differences. Fisher's exact tests were performed at a significance level of 5% to compare the findings of this study with those of Kerans et al. [15, 23] regarding methods mentioning, patient population, geography, clinical context, and treatment. Corresponding 95% confidence intervals (CI) were provided. Furthermore, overall tests were performed to compare frequencies of these items between all journals. Bias-corrected Cramérs V effect measures were estimated with corresponding parametric bootstrapped CIs. The DerSimonian and Laird [30] (DSL) approach was used to perform random effect (RE) meta-analyses, which allows for variability in the variables of interest properties between journals and over time. The logit transformation was used for estimating the pooled proportions [31], and standard errors were not back-transformed.

The effect of time regarding the specific title properties was investigated by logistic regression models, if appropriate. Post hoc comparisons for the identification of homogeneous subgroups were performed using Tukey's HSD. Associations between title properties and the journals were analyzed using likelihood ratio tests. Effect estimates, i.e., odds ratios and corresponding 95% CI were reported for all analyses, and the journal BMJ was used as reference category. An odds ratio of x.x being greater than 1 indicates an x.x fold increased chance containing the specific item for an one-year difference adjusted for the variable journal.

Data and R code for all analyses are provided in **S1 and S2 Files**, respectively.

## Results

A total of 500 randomly selected original research articles from 5 medical journals were analyzed regarding the selected title items (see **Table 1**). In **Table 2**, the descriptive statistics, i.e., absolute and relative frequencies for all title properties over the years are shown, respectively for each journal. Results of the meta-analyses are provided in detail in **S3 File, sections 4 and 5**.

### Items–Content

In terms of the title content topic methods, the NEJM deviated from the other journals regarding the methods mentioning. While methods were mentioned in at least 93% of the article titles in BMJ, Lancet and PLOS, about the half (47%) was in JAMA and 11% in NEJM article titles. Similar results were reported by Kerans et al. [15, 23] for BMJ, JAMA and Lancet, but proportions differed between Lancet titles (**Table 3**). The mentioning of methods increased over time (OR: 1.12 (95% CI: [1.01–1.24]), p = 0.025, **Fig 1** and **S3 File, section 6.1.1**), i.e., methods were mentioned more frequently in the article titles more recently.

Lowest and highest numbers for the mentioning of the patient population were in the BMJ (62%) and the NEJM (78%), respectively. For the mentioning of the patient population, neither an increase over time (OR: 1.06 (95% CI: [0.99–1.13]), p = 0.100, **Fig 1**) nor substantial differences between the journals (**S3 File, section 6.1.2**) could be observed.

About half of the PLOS titles (52%) contained any geographic information, but only 31% of the BMJ titles (see **Table 2**). Frequencies were only 16% and 17% for JAMA and Lancet, respectively, and 9% for NEJM titles. These findings are in line with Kerans et al. [15, 23], except for the BMJ, where Kerans et al. observed that 15.8% of the articles mentioned geographic information (**Table 3**). Mentioning of geographic information varied over time both within each journal (**S3 File, section 4.1.3.1**) and over the journals (**S3 File, section 4.1.3.2**). This is consistent with the results from the logistic regression analysis (OR: 1.07 (95% CI: [0.99–1.16]), p = 0.072, **Fig 1** and **S3 File, section 6.1.3**).

The clinical context was mentioned in 73% of BMJ titles, while it was mentioned at least 80% in the other four journals. This is in line with Kerans et al. [15, 23] (**Table 3**). Additionally, we observed an increase of clinical context mentioning over time (OR: 1.10 (95% CI: [1.01–1.19]), p = 0.025, **Fig 1** and **S3 File, section 6.1.4**).

**Table 2. Descriptive statistics for title properties of original articles for the period 2011 until 2020.** Absolute and relative frequencies (parenthesis) are shown.

| Title | Topic | Variable | BMJ | JAMA | Lancet | NEJM | PLOS |
|---|---|---|---|---|---|---|---|
| **Content** | **Methods** | Methods mention (yes) | 98 (98%) | 47 (47%) | 93 (93%) | 11 (11%) | 95 (96%) |
| | | Patient population (yes) | 62 (62%) | 76 (76%) | 72 (72%) | 78 (78%) | 70 (70%) |
| | | Geography (yes) | 31 (31%) | 16 (16%) | 17 (17%) | 9 (9%) | 52 (52%) |
| | | Clinical context (yes) | 73 (73%) | 81 (81%) | 83 (83%) | 89 (89%) | 81 (81%) |
| | | Treatment (yes) | 30 (30%) | 55 (55%) | 62 (62%) | 58 (58%) | 27 (27%) |
| | | Study name (yes) | 11 (11%) | 20 (20%) | 45 (45%) | 0 (0%) | 10 (10%) |
| | **Results** | Results mention (yes) | 1 (1%) | 0 (0%) | 1 (1%) | 2 (2%) | 2 (2%) |
| | | Quantitative information (yes) | 0 (0%) | 0 (0%) | 0 (0%) | 0 (0%) | 0 (0%) |
| | | Semi-quantitative information (yes) | 0 (0%) | 0 (0%) | 1 (1%) | 2 (2%) | 1 (1%) |
| | | Relation (yes) | 52 (52%) | 68 (68%) | 35 (35%) | 23 (23%) | 57 (57%) |
| **Form** | **Methods** | Abbreviation (yes) | 31 (31%) | 44 (44%) | 55 (55%) | 24 (24%) | 32 (32%) |
| | | Dash (yes) | 3 (3%) | 0 (0%) | 0 (0%) | 2 (2%) | 0 (0%) |
| | | Subtitle (yes) | 99 (99%) | 41 (41%) | 99 (99%) | 2 (2%) | 98 (98%) |
| | **Discussion/conclusion** | Declarative title (yes) | 0 (0%) | 0 (0%) | 0 (0%) | 0 (0%) | 0 (0%) |
| | | Phrasal title (yes) | 11 (11%) | 3 (3%) | 12 (12%) | 7 (7%) | 15 (15%) |
| | | Question (yes) | 1 (1%) | 0 (0%) | 1 (1%) | 0 (0%) | 1 (1%) |

JAMA: The Journal of the American Medical Association, NEJM: The New England Journal of Medicine, PLOS: PLOS Medicine.

Only 27% in PLOS and 30% in BMJ provided some treatment information in the title, while for the other three journals at least 50% of the article titles mentioned a treatment. Our results did not show any differences from those of Kerans et al. [15, 23] (**Table 3**). Over time the naming of treatments in the title increased (OR: 1.08 (95% CI: [1.02–1.16]), p = 0.015, **Fig 1** and **S3 File, section 6.1.5**).

There was no NEJM title containing a study name while Lancet had the highest usage of it (45%). The analysis over time showed a trend over time (OR: 1.13 (95% CI: [1.03–1.24]), p = 0.008) and substantial differences between the journals (**S3 File, section 6.1.6**).

Regarding the title topic results, only 6 out of the total of 500 articles mentioned results in their titles. This is in line with the findings of Kerans et al. [15, 23], who reported that 1.9% of NEJM titles mentioned results. No article provided any quantitative information in its title, and only 4 of 500 articles provided semi-quantitative information in their title. Because of very low numbers, no further analyses were performed for these criteria.

A relation between variables was used least frequently in the NEJM (23%), followed by the Lancet (35%). The other three journals mentioned a relation in more than half of the articles (**Table 2**). These differences between journals were confirmed in regression analysis (**S3 File, section 6.2.4**). However, an increase over time could not be observed (p = 0.858, **Fig 1**).

## Items–Form

In terms of the title form topic methods, abbreviations were less used in NEJM titles and most used in Lancet titles, 24% and 55 respectively (see **Table 2**). An increase use over time was observed (OR: 1.13 (95% CI: [1.05–1.20]), p < 0.001, **Fig 1**) as well as differences between journals (**S3 File, section 7.1.1**).

Dashes were rarely used. Only three articles in BMJ and two articles in NEJM used a dash (**Table 2**). Further analyses were not performed because of these low frequencies.

A subtitle was used in at least 98% of the articles in BMJ, Lancet, and PLOS, while only 41% of JAMA titles and only 2% of NEJM titles used subtitles. These clear differences between the

**Table 3. Relative frequencies (relFreq) of title properties (yes) with results from Kerans et al. [15, 23] and this study.** Corresponding 95% confidence intervals (CI) are shown in brackets. Results of PLOS Medicine are missing because Kerans et al. did not examine article titles of this journal.

| | BMJ | | JAMA | | Lancet | | NEJM | | | |
|---|---|---|---|---|---|---|---|---|---|---|
| Variable | relFreq [95% CI] | p[1] | relFreq [95% CI] | p[1] | relFreq [95% CI] | p[1] | relFreq [95% CI] | p[1] | V[2] [95% CI] | p[3] |
| **Methods mention** | | | | | | | | | | |
| Kerans et al. | 98.2 [89.0–99.9] | 1.000 | 45.9 [33.3–59.1] | 1.000 | 97.4 [90.0–99.5] | 0.303 | 11.3 [4.7–23.7] | 1.000 | 0.75 [0.67–0.82] | <0.001 |
| This study | 98.0 [92.3–99.7] | | 47.0 [37.0–57.2] | | 93.0 [85.6–96.9] | | 11.0 [5.9–19.2] | | 0.73 [0.68–0.79] | <0.001 |
| **Patient population** | | | | | | | | | | |
| Kerans et al. | 27.6 [18.3–39.3] | <0.001 | 29.5 [18.9–42.7] | <0.001 | 19.7 [11.8–30.8] | <0.001 | 26.2 [16.2–39.3] | <0.001 | 0 [0–0.06] | 0.560 |
| This study | 62.0 [51.7–71.4] | | 76.0 [66.2–83.7] | | 72.0 [62.0–80.3] | | 78.0 [68.4–85.4] | | 0.11 [0–0.22] | 0.065 |
| **Geography** | | | | | | | | | | |
| Kerans et al. | 15.8 [8.8–26.4] | 0.022 | 14.8 [7.4–26.7] | 1.000 | 21.1 [12.9–32.2] | 0.560 | 14.8 [7.4–26.7] | 0.446 | 0 [0–0.06] | 0.724 |
| This study | 31.0 [22.3–41.1] | | 16.0 [9.7–25.0] | | 17.0 [10.5–26.1] | | 9.0 [4.5–16.8] | | 0.18 [0.04–0.28] | 0.002 |
| **Clinical context** | | | | | | | | | | |
| Kerans et al. | 80.3 [69.2–88.2] | 0.289 | 82 [69.6–90.2] | 1.000 | 82.9 [72.2–90.2] | 1.000 | 90.2 [79.1–95.9] | 1.000 | 0 [0–0.04] | 0.431 |
| This study | 73.0 [63.0–81.2] | | 81.0 [71.7–87.9] | | 83.0 [73.9–89.5] | | 89.0 [80.8–94.1] | | 0.12 [0–0.22] | 0.036 |
| **Treatment** | | | | | | | | | | |
| Kerans et al. | 36.8 [26.3–48.7] | 0.418 | 44.3 [31.8–57.5] | 0.198 | 72.4 [60.7–81.7] | 0.197 | 57.4 [44.1–69.7] | 1.000 | 0.29 [0.17–0.38] | <0.001 |
| This study | 30.0 [21.5–40.1] | | 55.0 [44.8–64.9] | | 62.0 [51.7–71.4] | | 58.0 [47.7–67.7] | | 0.24 [0.13–0.33] | <0.001 |

JAMA: The Journal of the American Medical Association, NEJM: The New England Journal of Medicine. [1]p-values from Fisher exact test; frequencies were compared within a journal for the respective variable. [2]Cramérs V (bias-corrected); CIs calculated by bootstrapping (normal approximation, 100 replications), [3]p-values from Fisher-Freeman-Halton exact test; frequencies were compared between all journals within the respective study

journals were confirmed with the regression analysis (**S3 File**, **section 7.1.2**). Moreover, the usage of subtitles increased over time (OR: 1.22 (95% CI: [1.07–1.38]), p < 0.003, **Fig 1**).

Finally, regarding the title form topic discussion, not a single article had a declarative title in our analyses (**Table 2**). Phrasal titles were present in 3% of JAMA, 7% of NEJM, 11% of BMJ, 12% of Lancet, and 15% of PLOS titles. Significant differences between journals could not be observed (**S3 File**, **section 7.2.2**). A decrease of phrasal titles over time was observed in the regression analysis (OR: 0.90 (95% CI: [0.81–1.00]), p < 0.044, **Fig 1** and **S3 File**, **section 7.2.2**).

Only three of 500 article titles were written as a question (**Table 2**). Kerans et al. [15, 23] observed similar low frequencies; and they reported 3.9% for the BMJ and 1.3% for Lancet articles with a question symbol, and none for both JAMA and NEJM (**Table 3**).

## Geographic information–Manual versus automated search with the `maps` package

The comparison of our hand search on the mentioning of geographic information revealed substantial differences to the automated search with the R package `maps` [9].

In detail, respectively, 31% vs. 13% for BMJ, 16% vs. 3% for JAMA, 17% vs. 9% for the Lancet, 9% vs. 3% for the NEJM and 52% vs. 29% for PLOS articles contained any geographical information in their titles for the hand and automatic search. The automated search thus led to fewer titles with any geographic information.

## Discussion

Title content properties varied substantially between original research articles published in the general major medical journals. Furthermore, title content and form changed over time.

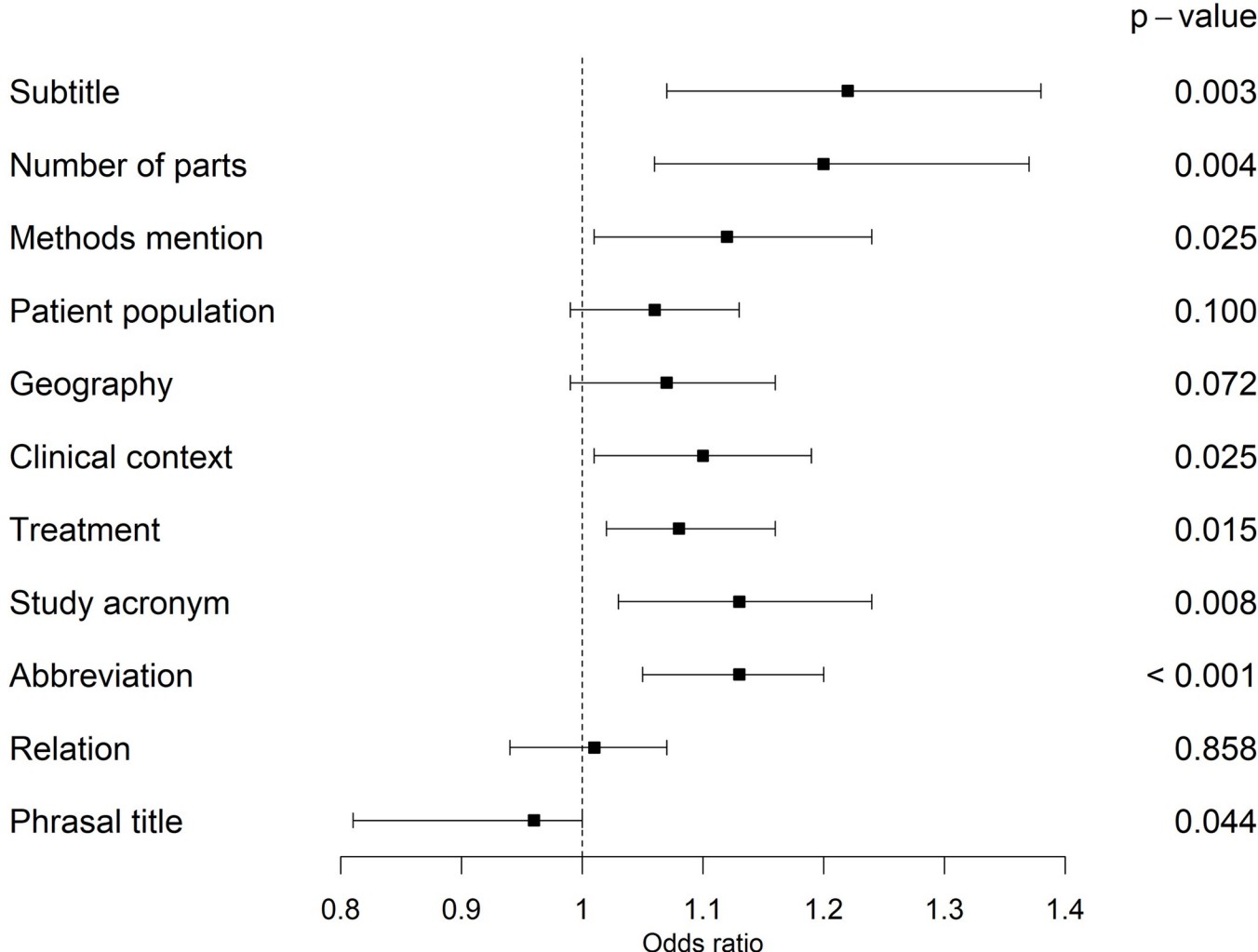

**Fig 1. Forest plots for the effect of time on title content and form.** Displayed are odds ratios (square) per increase by one year, corresponding 95% confidence intervals (whisker) and p-values (numbers).

Differences between journals were specifically observed in the use of subtitles. While almost all articles from the BMJ and PLOS had subtitles, only two of the NEJM articles had a subtitle. Previously, we and others showed that the colon was most used in titles to split a title into multiple parts rather than any other separator [9, 15, 23]. Here, we furthermore showed that the proportion of paper with subtitles increased over time.

Substantial differences between journals were also observed for the mentioning of methods, the patient population, the geography, the interventional treatment, and the use of an abbreviation in the title. In addition, there were substantial differences in the use of a study name in the title. For example, while no article published in the NEJM used a study name, almost half (45%) of the studies in the Lancet used one. Some content criteria were mainly not or rarely used in all considered journals, such as a dash, mentioning of results, using a declarative title, or a question mark. This was in contrast to Paiva, Lima [32] who showed for PLOS and BMC journals that approximately 40% of the articles mentioned the results, and such articles were more frequently cited than work mentioning methods. In our study, only 6 articles out of 500 mentioned results in the title, while 344 out of the 500 articles mentioned of methods. Our

findings are in line with general guidelines that declamatory titles, i.e., titles that give study results should be avoided [33]; see, e.g., instructions to authors for the Lancet. Authors should thus avoid providing quantitative or semi-quantitative information in the title. In fact, since the title is a one-line summary, the conclusions could be spread out into the world without reading at least the abstract or the full text of the article. Aleixandre-Benavent and colleagues go a step further and provide recommendations what a title should contain, and how it should not be constructed [16].

Our work focused on the general major medical journals plus the online only journal PLOS. Between the printed journals, there were substantial differences regarding the content of article titles [9]. One of the reasons could be in the instructions for authors, which differ in the provided information on the construction of a title. Specifically, the NEJM title had the lowest number of frequencies for a couple of criteria, such as the subtitle, methods mentioning, geography, abbreviations, and relation. No NEJM title contained a study name. However, the clinical context and the patient population was most frequently described in NEJM article titles. Differences between printed and online journals were obvious using geographic information in the title or usage of a phrasal title occurring more often in the online journal PLOS.

Subtitles are now more frequently used than a decade ago. Furthermore, the mentioning of methods increased in the 10 years from 2011 to 2020. This change in the title may be caused by the increased use of reporting guidelines, such as the CONSORT statement [34], which states that a randomized controlled trial should be identifiable as randomized in the title. The instructions for authors of all considered journals state that subtitles should be used for reporting the study design and/or authors should follow the respective reporting guidelines of their study. In fact, authors should look out a copy of the target journal and identify its preferences [35].

Our results are in line with the recommendations from the journal-specific instructions for authors, except NEJM. The NEJM does not follow the CONSORT statement using subtitles for randomized controlled trials, see also [1]. For the other four journals, the mentioning of the study design or the type of analysis is almost always done using subtitles as recommended. Furthermore, our results for JAMA using no declarative titles, no results mentioning or using questions in the title match with its recommendations.

Research has so far concentrated on the form of article titles rather than its content. While some authors investigated title content in BMJ, JAMA, Lancet and NEJM for a specific time, generally a single year [15, 23, 36], the development of title content over time has rarely been studied [37]. A strength of our work thus is the availability of all original articles over a time span of 10 years [9]. From this database, we randomly selected a subset of articles for manual assessment. These articles were evaluated by two raters according to a pre-specified coding plan with examples and training. Title evaluations were then done blinded by year and journal.

We did not expect different journal-specific frequencies regarding the geographic information in the title compared to our recent work [9], in which we performed an automatic search for country and city mentioning in the title by the use of the R package `maps` [9]. However, frequencies differed substantially. The automated search led to fewer titles with any geographic information. For example, the `maps` package did not contain countries, such as 'England', continents, abbreviation, such as 'U.S.', or terms, such as 'English'. The main reasons for the discrepancies were for the use of country-specific abbreviations and additional country-specific terms. However, other tools or packages might have been more appropriate for the geographical query than the `maps` package.

One limitation of our study is that we relied on the quality of the data provided by the PubMed database [38]. Another limitation of our work is that additional variables could have been considered, e.g., more complex title content [12, 16, 22].

A further limitation is the sample size of 500 articles, i.e., 10 articles per journal and year. With a sample size substantially larger than 1000 articles we would have been able to study the association of title characteristics with citation counts. For example, the total sample size of our previous study, which was based on an automated search was 8096 articles [9]. With 500 articles, 95% confidence intervals are approximately 4 times larger ($\sqrt{8096} / \sqrt{500} = 4.02$), and many results, such as the association between the number of citations would not have been significant. The sample size used in this study is approximately twice that of [15, 23], and this study with 500 articles was powered to reliably detect trends over time.

In future research, it would be of interest to analyze the effect of title content properties on citation frequencies. It would also be interesting to compare specific journals with general medical journals.

In conclusion, title content differed substantially between the five major medical journals BMJ, JAMA, Lancet, NEJM and PLOS. Furthermore, title content changed over time. We recommend that authors study titles of articles recently published in their target journal when formulating the manuscript title. Analyses of title content may generally require manual time-consuming inspections.

## Supporting information

**S1 File. Data.**
(XLSX)

**S2 File. R markdown file for analyses.**
(RMD)

**S3 File. Results.**
(HTML)

## Author Contributions

**Conceptualization:** Nicole Heßler, Andreas Ziegler.

**Data curation:** Nicole Heßler.

**Formal analysis:** Nicole Heßler, Andreas Ziegler.

**Investigation:** Nicole Heßler, Andreas Ziegler.

**Methodology:** Nicole Heßler, Andreas Ziegler.

**Project administration:** Nicole Heßler, Andreas Ziegler.

**Resources:** Nicole Heßler, Andreas Ziegler.

**Software:** Nicole Heßler.

**Supervision:** Andreas Ziegler.

**Validation:** Nicole Heßler, Andreas Ziegler.

**Visualization:** Nicole Heßler, Andreas Ziegler.

**Writing – original draft:** Nicole Heßler, Andreas Ziegler.

**Writing – review & editing:** Nicole Heßler, Andreas Ziegler.

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
