## [Decision Letter · Decision Letter 0]

10 Apr 2023

PONE-D-23-07021Title Content and Form of Original Research Articles in High-Ranked Medical JournalsPLOS ONE

Dear Dr. Ziegler,

Thank you for submitting your manuscript to PLOS ONE. After careful consideration, we feel that it has merit but does not fully meet PLOS ONE’s publication criteria as it currently stands. Therefore, we invite you to submit a revised version of the manuscript that addresses the points raised during the review process.

We look forward to receiving your revised manuscript.

Kind regards,

Boyen Huang, DDS, MHA, PhD

Academic Editor

PLOS ONE

Journal Requirements:

   "There are no patents, products in development or marketed products to declare. AZ is a licensed Tim Albert trainer and has held several courses in the past based on Albert’s concept."

Additional Editor Comments:

The major concerns from the reviewers include the sample size, sample selection, and writing style.

Reviewers' comments:

Reviewer's Responses to Questions

**Comments to the Author**

1. Is the manuscript technically sound, and do the data support the conclusions?

Reviewer #1: Yes

Reviewer #2: Yes

Reviewer #3: Partly

Reviewer #4: Yes

2. Has the statistical analysis been performed appropriately and rigorously? 

Reviewer #1: I Don't Know

Reviewer #2: Yes

Reviewer #3: Yes

Reviewer #4: I Don't Know

3. Have the authors made all data underlying the findings in their manuscript fully available?

Reviewer #1: Yes

Reviewer #2: Yes

Reviewer #3: No

Reviewer #4: Yes

4. Is the manuscript presented in an intelligible fashion and written in standard English?

Reviewer #1: No

Reviewer #2: Yes

Reviewer #3: Yes

Reviewer #4: Yes

5. Review Comments to the Author

Reviewer #1: This article analyses the titles of articles published in a series of medical journals over time. It is interesting for a consideration of how naming practices affect discoverability and use of research material.

There are, though, a few aspects that need revision:

First, the title of this paper makes reference to “high ranked” medical journals, but there is no definition anywhere of how this ranking is constructed or to what ranking you are referring.

Second, the sample size of 500 is relatively small and limits the general applicability of the findings, as the paper notes. I am unclear as to whether this is too small to be useful/generalizable.

Third, some of the limitations could have been overcome with different computational methods. For example, on page 14 you state that the maps package that you used was not able correctly to identify many locations in article headings. However, other named-entity recognition tools would certainly do a better job of this. For instance, Amazon Comprehend or SageMaker could be appropriate tools here.

Fourth, the language needs careful checking throughout. For instance: “can only adequately [be] measured”; “articles meaning no sentence and no question”; “we did neither observe” → “we observed neither”; “almost the half” → “almost half”.

Fifth, and perhaps most significantly, it would be helpful for the conclusions of this paper to interpret the findings more closely. Why have these changes that you find occurred? What does it mean that subtitles are now more common? How does discoverability work in each of the title types to which you refer?

Finally, you open by stating that the prime driver of picking a good title is so that you can pick up citations and have career progression. This seems a very cynical way of thinking about how to title articles. Scientists and medics should use titles that accurately reflect the content of the work and allow others easily to find and re-use their research. I would suggest amending this opening to incorporate such a stance.

Reviewer #2: The paper titled “Title Content and Form of Original Research Articles in High-Ranked Medical Journals” investigates the differences of title content and form between papers in the medical field and their changes over time. Overall, the paper is well-written and well-argued. The methodology is adequate, and there are an overall coherence and relation to the scope of publication in the Plos One journal. In addition, this manuscript addresses a very interesting issue about the analysis of titles and does so in a very competent technical way. It Is worth mentioning that data and R code are shared.

I do however have a major issue (which in fact, is a minor one). The authors did a huge effort in sharing all the data; however, the results section sometimes it's difficult to follow (the reader should go back and forth checking the tables). I think some (introductory) sentences in some parts of the manuscript will benefit the readability of the text (see my suggestions below).

I will go slightly more into detail with them in the position-specific comments below.

Keywords:

I think the paper would benefit by including some keywords related to the titles (e.g. research article titles or titles). I have reservations about the use of ‘impact’ (I think the authors are not analysing the impact of the papers, e.g. citation impact).

Introduction

The introduction and background is ok, providing the necessary information leading to the purpose of the study. However, I think the authors could expand a bit more (there are more studies on the topic). See my suggestions below.

P10|Line 56. Indicate that the EQUATOR Network is referred to the reporting health research (e.g. Enhancing the QUAlity and Transparency Of health Research (EQUATOR) Network).

P10|Line 33 (and P11|Line 83). “content has rarely been investigated beyond title length”. I slightly disagree with this statement. From a bibliometric perspective, there are many articles that analyse impact (e.g. effect on citations, downloads), sentence types (e.g. informative), the information the author wants to include, and in which order, among others. I would like to suggest the following papers (not included by the authors):

• Aleixandre-Benavent, R., Montalt-Resurecció, V., & Valderrama-Zurián, J. C. (2014). A descriptive study of inaccuracy in article titles on bibliometrics published in biomedical journals. Scientometrics, 101(1), 781–791. https://doi.org/10.1007/s11192-014-1296-5.

• Ball, R. (2009). Scholarly communication in transition: The use of question marks in the titles of scientific articles in medicine, life sciences and physics 1966–2005. Scientometrics, 79(3), 667–679. https://doi.org/10.1007/s11192-007-1984-5.

• Busch-Lauer, I.-A. (2000). Titles of English and German research papers in medicine and linguistics theses and research articles. In A. Trosborg (Ed.), Analysing professional genres (pp. 77–94). John Benjamins Publishing Company. https://doi.org/10.1075/pbns.74.08bus

• Buter, R. K., & van Raan, A. F. J. (2011). Non-alphanumeric characters in titles of scientific publications: An analysis of their occurrence and correlation with citation impact. Journal of Informetrics, 5(4), 608–617. https://doi.org/10.1016/j.joi.2011.05.008.

• Haggan, M. (2004). Research paper titles in literature, linguistics and science: Dimensions of attraction. Journal of Pragmatics, 36(2), 293–317. https://doi.org/10.1016/S0378-2166(03)00090-0

• Nagano, R. L. (2015). Research article titles and disciplinary conventions: A corpus study of eight disciplines. Journal of Academic Writing, 5(1), 133–144. https://doi.org/10.18552/joaw.v5i1.168

• Pearson, W. S. (2021). Quoted speech in linguistics research article titles: patterns of use and effects on citations. Scientometrics, 126(4), 3421-3442.

P10|Line 75. Worth mentioning the Web of Science (1997) which includes the title field tag.

P11|Line 79. Indicate the acronym (Standards for Reporting of Diagnostic Accuracy (STARD)). When the authors mention ‘at least two observers should do an independent evaluation where applicable”. Are referring to the article title? (not clear)

P11|Line 84. Correct typo “(2020)compared”.

Methods

P12|106-108. The authors mention “Articles being listed with an abstract remained in the data set, while articles only listed in the Web of Science were excluded.”. I suggest indicating the number of papers. Was the abstract used for any purpose?

P12|111. Indicate in this section that ten original articles per year (100 articles per journal) were randomly chosen.

Results

There is a lot of information in this section (Tables and supplementary material), which allows the reproducibility of the findings. However, I think some introductory sentences will benefit the readability of the text (see my suggestions below).

P14|159-160. Although the information is in the Supplementary Material, I suggest introducing a few words (just one or two sentences) about Table 2 (or Descriptive Statistics).

In Table 3, the Plos Medicine journal information is missing in the table.

P14|159-160 “About half of the PLOS titles (52%) contained any geographic information”: missing this information in Table 3 (or indicate the Supplementary table in which this information is displayed).

P16|203. Correct typo ‘ofKerans’.

P16|207. Here, the authors mention the Results/Relation (and not the previous ones, i.e. Results mention, Quantitative information or Semi-quantitative information). I suggest an introductory sentence indicating that ‘In terms of results, etc’. And also pointing out that the other previous items were rarely used.

P16|214. Regarding this information (24%), indicate in brackets Table 2 (or 3.2. Supplementary Table)

P16|224. Indicate that refers to the discussion/conclusion part.

Discussion/Conclusions

Line 19|278-279. Another aspect that should be considered is the title length allowed by each journal (number maximum of words). Also there is of interest the recommendations from the journals (e.g. in the author guidelines). In some journals there is some criteria such as ‘Specific, descriptive, concise, and comprehensible to readers outside the field’ (Plos One), whereas in others it is suggested to include a subtitle (e.g. https://jamanetwork.com/journals/jamanetworkopen/pages/instructions-for-authors).

Line 20 |321. Another limitation is the variables considered (e.g. some other studies analyse other non-alphanumeric elements such as exclamations, other criteria for the content, etc.)

Line21|333. A sentence about further research could be included.

Reviewer #3: • There was a desire to look at characteristics within journals over time but use of only 10 articles per year seems subject to selection bias for this research question. What was the power consideration here?

• It is not clear how the titles were evaluated. Was it an automated program or each one manually? The samples size is small enough that manual adjudication is possible.

• Line 115. This is confusing. Results are not given for harmonization of classification of various title attributes.

• Reporting of ORs is confusing. For example on line 170, what is the OR for? The proportion listing method per year? Similar in line 188 – you say the rate varied over time, but you used logistic regression assuming an increase over time?

• The results section includes discussion points (like line 202).

• Were any findings linked back to author instructions for each journal? These often dictate title content.

• Linking the metrics assessed to citation counts would add an important dimension to the significance of this research.

• There is too much repetition of p values in the Discussion. I assume these were not presenting new analyses not shown in the results. It is not appropriate. Line 312 – they do report new analyses. It should be part of the study of not (unless published elsewhere).

• The Discussion is too long.

• The recommendation near the end that authors study titles in their target journals before submission is unfounded. Title is often dictated by author guidelines or changed during peer review. They did not study this particular question – in other words, they did not study title of rejected compared to accepted articles.

Reviewer #4: This study applies what are in my opinion very sound methodologies to analyze the titles of prestigious, general medical journals. The paper is well-written, and its significance lays on going beyond other studies in investigating titles’ form & content and the development of title content over time. To do so, they had to select a representative sample of articles over a period of 10 years. Having two (trained) raters made the methodology strong, as it was the methodology followed in the “Evaluation of title content and form” section. (I have to admit, however, that I lack the expertise to say that the statistical analyses have been performed appropriately and rigorously. So in the following I assume these have been done correctly.) By following this well-crafted methodology, and providing all the relevant data, the code to analyze it, and in detail results in the supplementary materials, the conclusions arrived at are well supported (see some comments below, though) and could be replicated by others.

I do have some specific concerns or comments that I would like the authors to address:

1. I think the authors should stick to the wording “general major medical journals” instead of “highly ranked” as they don’t define which “rank” that is or where it can be found or calculated.

2. Mentioning of guidelines for authors writing the papers in the journals analyzed was not mentioned at all ---even as it is mentioned in the literature they quote. I think this is important as to it may be determining why authors use a particular way to phrase the title. The reader is left to assume that no guidance was provided by the journal that could have biased title wording. I think this to be particularly important for the use or avoidance of abbreviations, dashes, and/or subtitles.

3. The authors recommendation “We recommend that authors study titles of articles recently published in their target journal when formulating the manuscript title” does not seem supported other than by their results implying this is what you find in them already. So, why should you follow the same? Would that make it more likely to be published? The paper’s introduction makes reference to increasing citation frequency in databases, and so does at least one of the authors’ previous papers, yet it’s never mentioned explicitly as a possible outcome of choosing title according to the journal to submit.

4. Regarding their recommendation “In our opinion, authors should avoid providing quantitative or semi-quantitative information in the title. In fact, since the title is a one-line summary, the conclusions could be spread out into the world without reading at least the abstract or the full text of the article. ”I think this argument should expand as to what the consequences are in following this behavior e.g. propagation of misinformation.

5. In their statement “Another limitation of our study is that we relied on the quality of the data provided by the database of PubMed. Specifically, we may have missed some original articles in our database search. And we have previously identified a couple of errors in the database (Heßler and Ziegler, 2022).” One shouldn’t expect the reader to go their paper for finding out what was wrong with those hits/articles.

6. Finally, author AZ declares, in the competing interests field, that he's a "licensed Tim Albert trainer and has held several courses in the past based on Albert’s concept." Please consider adding the statement that (at least some of the) Tim Albert trainings deal with advising people how to write medical papers.

P.S. There are a few typos, like missing words and letters, that need to be corrected throughout the manuscript.

6. PLOS authors have the option to publish the peer review history of their article (what does this mean?). If published, this will include your full peer review and any attached files.

Reviewer #1: No

Reviewer #2: No

Reviewer #3: No

Reviewer #4: No

---

## [Decision Letter · Decision Letter 1]

12 Jun 2023

Title Content and Form of Original Research Articles in General Major Medical Journals

PONE-D-23-07021R1

Dear Dr. Ziegler,

We’re pleased to inform you that your manuscript has been judged scientifically suitable for publication and will be formally accepted for publication once it meets all outstanding technical requirements.

Kind regards,

Boyen Huang, DDS, MHA, PhD

Academic Editor

PLOS ONE

Additional Editor Comments (optional):

Reviewers' comments:

Reviewer's Responses to Questions

**Comments to the Author**

1. If the authors have adequately addressed your comments raised in a previous round of review and you feel that this manuscript is now acceptable for publication, you may indicate that here to bypass the “Comments to the Author” section, enter your conflict of interest statement in the “Confidential to Editor” section, and submit your "Accept" recommendation.

Reviewer #2: All comments have been addressed

Reviewer #4: All comments have been addressed

2. Is the manuscript technically sound, and do the data support the conclusions?

Reviewer #2: Yes

Reviewer #4: (No Response)

3. Has the statistical analysis been performed appropriately and rigorously? 

Reviewer #2: Yes

Reviewer #4: (No Response)

4. Have the authors made all data underlying the findings in their manuscript fully available?

Reviewer #2: Yes

Reviewer #4: (No Response)

5. Is the manuscript presented in an intelligible fashion and written in standard English?

Reviewer #2: Yes

Reviewer #4: (No Response)

6. Review Comments to the Author

Reviewer #2: I have read the second revision of this manuscript and, in my opinion, the authors have addressed the main issues and provided a satisfactory answer in their response.

Reviewer #4: (No Response)

7. PLOS authors have the option to publish the peer review history of their article (what does this mean?). If published, this will include your full peer review and any attached files.

Reviewer #2: No

Reviewer #4: No

---

## [Editor Report · Acceptance letter]

15 Jun 2023

PONE-D-23-07021R1 

Title Content and Form of Original Research Articles in General Major Medical Journals 

Dear Dr. Ziegler:

I'm pleased to inform you that your manuscript has been deemed suitable for publication in PLOS ONE. Congratulations! Your manuscript is now with our production department. 

Kind regards, 

on behalf of

Dr Boyen Huang 

Academic Editor

PLOS ONE